Cranial morphology of Sinovenator changii (Theropoda: Troodontidae) on the new material from the Yixian Formation of western Liaoning, China

Yin Ya-Lei 1
Pei Rui 2 peirui@hku.hk
Zhou Chang-Fu 3
1 Paleontological Institute, Shenyang Normal University , Shenyang, Liaoning , China
2 Department of Earth Sciences, the University of Hong Kong , Hong Kong , China
3 College of Earth Science and Engineering, Shandong University of Science and Technology , Qingdao, Shandong , China
Sues Hans-Dieter
Electronic publication date: 2018 Jun 20
Publication date: 2018
Volume: 6
Electronic Location ID: e4977
Received 2018 Mar 26; Accepted 2018 May 24
Copyright: © 2018 Yin et al.
Copyright year: 2018
Copyright holder: Yin et al.
License: This is an open access article distributed under the terms of the Creative Commons Attribution License, which permits unrestricted use, distribution, reproduction and adaptation in any medium and for any purpose provided that it is properly attributed. For attribution, the original author(s), title, publication source (PeerJ) and either DOI or URL of the article must be cited.
License URL: https://creativecommons.org/licenses/by/4.0/

Keywords: Troodontidae, Sinovenator, Jehol Biota, Yixian Formation, Early Cretaceous

Funding: Shandong Provincial Natural Science Foundation ZR2017MD031 Liaoning BaiQianWan Talents Program 2014Q110 This project was supported by Shandong Provincial Natural Science Foundation (ZR2017MD031) and Liaoning BaiQianWan Talents Program (No.2014Q110). The funders had no role in study design, data collection and analysis, decision to publish, or preparation of the manuscript.

==============================
A new three-dimensionally preserved troodontid specimen consisting of most of the skull, partial mandibles and six articulated cervical vertebrae (PMOL-AD00102) from the Early Cretaceous Yixian Formation of Beipiao, western Liaoning, China is identified as Sinovenator changii on the basis of a surangular with a “T”-shaped cross-section. High-resolution computed tomographic data for the skull of this new specimen facilitated a detailed description of the cranial anatomy of S. changii. New diagnostic features of S. changii include a well-developed medial shelf on the jugal, a slender bar in the parasphenoid recess, a lateral groove on the pterygoid flange of the ectopterygoid, and the lateral surface of the anterior cervical vertebrae bearing two pneumatic foramina. Our new observation confirms that the braincase of Sinovenator is not as primitive as previously suggested, although it still shows an intermediate state between derived troodontids and non-troodontid paravians in having an initial stage of the subotic recess and the otosphenoidal crest. Additionally, this new specimen reveals some novel and valuable anatomical information of troodontids regarding the quadrate-quadratojugal articulation, the stapes, the epipterygoid and the atlantal ribs.

Introduction

Troodontidae is a group of small to middle-bodied theropod dinosaurs, and is well known from the Cretaceous rocks of Asia and North America (Makovicky & Norell, 2004). It has a high morphological relevance in understanding the avian origin (Xu et al., 2002). Many exquisitely preserved troodontid fossils have been reported from the Early Cretaceous Jehol Biota in western Liaoning and adjacent areas in the last two decades, such as Sinovenator, Mei, Sinusonasus, Jinfengopteryx, Daliansaurus, Liaoningvenator, and Jianianhualong (Xu et al., 2002, 2017; Xu & Norell, 2004; Xu & Wang, 2004; Ji et al., 2005; Shen et al., 2017a, 2017b). These discoveries shed new lights on the evolution of troodontids and the origin of birds (Xu et al., 2002, 2017; Xu & Norell, 2004). Among these recently reported troodontids, Sinovenator, with similarities to both troodontids and dromaeosaurids, has been believed to be one of the most basal members of Troodontidae, and plays a key role in understanding the origin and the early evolution of this family (Xu et al., 2002). However, only a few specimens of Sinovenator have been described in detail, including the two specimens (IVPP V12615 and IVPP V12583) reported in the original paper (Xu et al., 2002). The morphology of the snout and the braincase of Sinovenator changii have been carefully described based on the holotype in previous studies (Xu et al., 2002; Xu, 2002), however, the anatomical details of the middle of the posterior portions of the cranium are still lacking. Here, we report a new specimen of S. changii (PMOL-AD00102), discovered from the lowest part of the Yixian Formation at the Lujiatun locality of western Liaoning, China (Fig. 1). This fossil is comprised of a nearly complete skull, partial mandibles and six articulated cervical vertebrae. This new specimen is referred to Sinovenator changii based on proposed diagnostic characters of this species, such as a “T”-shaped cross-section of the surangular. In this study, we also employed high-resolution computed tomographic (CT) technology to reveal the cranial anatomy of PMOL-AD00102 that is still concealed in the matrix. The new anatomical information not only enriches our knowledge of the osteology of Sinovenator, but also provides an opportunity to investigate the evolutionary trends in the palate and cranium of troodontids.

Figure 1 Area map showing the fossil locality (marked by an asterisk) of Sinovenator (PMOL-AD00102) in Lujiatun Village, Shangyuan, Beipiao City, western Liaoning Province, China.

Materials and Methods

PMOL-AD00102 is preserved in three dimensions with a nearly complete skull, partial mandibles and six articulated cervical vertebrae (Figs. 2–15). The skull lacks the rostral portion anterior to the antorbital fenestra and is slightly anterolaterally compressed. The mandibles lack the rostral portions anterior to the last fourth dentary tooth. The specimen represents an adult individual as the neural arch and centrum of each cervical vertebra are fused.

The skull, mandibles and two articulated cervicals of PMOL-AD00102 (Figs. 2–13 and 15) were scanned by High-resolution X-ray CT scanner (Nikon XT H 320 LC; Nikon, Tokyo, Japan) at China University of Geosciences (Beijing), with a slice thickness of 50 μm at 90 kV and 274 μA. The dataset is comprised of 3000 DICOM files. Three-dimensional visualization and viewing on image slices were done using VG Studio Max 2.2 (Volume Graphics, Heidelberg, Germany).

Systematic Paleontology

Theropoda Marsh, 1881

Maniraptora Gauthier, 1986

Troodontidae Gilmore, 1924

Sinovenator changii Xu et al., 2002

Holotype

IVPP V12615 (Institute of Vertebrate Paleontology and Paleoanthropology), a partial skull and skeleton.

Paratype

IVPP V12583, an articulated partial postcranial skeleton.

Referred specimen

PMOL-AD00102 (Paleontological Museum of Liaoning), a partial skull and mandibles missing only the rostral portions, and six articulated cervical vertebrae (Figs. 2–15).

Locality, horizon and age

Lujiatun, Shangyuan, Beipiao City, western Liaoning, China (Fig. 1); the lowest part of the Yixian Formation, ca. 126 Ma (Chang et al., 2017). This specimen was collected from villagers at Lujiatun. The exact location where this specimen was discovered is unknown, but the greyish tuffaceous matrix and the three-dimensionally preserved skeleton strongly indicate that PMOL-AD00102 was from the tuffaceous fossil bed at the lowest part of the Yixian Formation, the major outcrop of which is located at Lujiatun.

Revised diagnosis

Sinovenator is distinguished from other troodontids in having the following autapomorphies (newly added diagnostic features marked by*): well-developed medial shelf on the jugal*; slender bar in the parasphenoid recess*; lateral groove on the pterygoid flange of the ectopterygoid*; surangular “T”-shaped in cross-section; lateral surface of the anterior cervical vertebrae bearing two pneumatic foramina*; and prominent lateral cnemial crest continuous with the fibular crest.

Description

Skull

The skull preserves partial antorbital fenestrae, large orbits and temporal fenestrae (Figs. 2–4). The preserved portion of the skull is about 78 mm long along the buccal margin from the anteroventral corner of the antorbital fenestra to the distal end of the articular joint. The antorbital fenestra is probably sub-rectangular as in Sinusonasus (see Figs. 1 and 2 in Xu & Wang, 2004), and the anterior margin of the antorbital fenestra is not complete. The anteroposterior length of the ventral margin of the antorbital fenestra is about 25 mm, larger than that of the holotype (IVPP V12615, 14 mm; Xu, 2002). The orbit is circular with a maximum diameter of about 40 mm.

Figure 2 Skull and mandibles of PMOL-AD00102 in left lateral view.

(A) photograph; (B) CT-rendered image. Study sites: an, angular; ax, axis; co, coronoid; cp, cultriform process; cr, cervical ribs; d, dentary; e, epipterygoid; f, frontal; j, jugal; l, lacrimal; ls, laterosphenoid; m, maxilla; n, nasal; p, parietal; pl, palatine; po, postorbital; pop, paroccipital process; pra, prearticular; q, quadrate; qj, quadratojugal; sd, supradentary; sp, splenial; sq, squamosal; su, surangular; v, vomer.

Figure 3 Skull and mandibles of PMOL-AD00102 in right lateral view.

(A) photograph; (B) CT-rendered image. Study sites: an, angular; atic, atlantal intercentrum; atna, atlantal neural arch; ax, axis; cp, cultriform process; d, dentary; ec, ectopterygoid; f, frontal; j, jugal; l, lacrimal; ls, laterosphenoid; m, maxilla; n, nasal; oc, occipital condyle; p, parietal; pl, palatine; po, postorbital; pop, paroccipital process; pra, prearticular; pro, proatlas; pt, pterygoid; q, quadrate; sp, splenial; sq, squamosal; su, surangular; v, vomer.

Figure 4 CT-rendered skull of PMOL-AD00102 in dorsal (A) and ventral (B) views.

Study sites: bpt, basipterygoid process; ec, ectopterygoid; f, frontal; j, jugal; l, lacrimal; ls, laterosphenoid; m, maxilla; n, nasal; nc, nuchal crest; oc, occipital condyle; p, parietal; pl, palatine; po, postorbital; pop, paroccipital process; pt, pterygoid; q, quadrate; qj, quadratojugal; rf, ridge on frontal; s?, a possible stapes fragment; sc, saggital crest; sq, squamosal; v, vomer.

Maxilla

Both maxillae are partially preserved (Figs. 2–4). A fragmentary ascending process of the maxilla is preserved on the right side of the skull. Laterally, it has a tapering tip and contacts the anterior process of the lacrimal, both forming the dorsal margin of the antorbital fenestra (Figs. 3 and 4B). Only the ventral portion of the interfenestral bar is preserved on the right side of the skull, and the interfenestral bar appears to be vertical, as in the holotype (Xu et al., 2002) and Sinusonasus (Xu & Wang, 2004), forming the anterior margin of the antorbital fenestra. The ventral ramus (jugal process) of the maxilla is slender, as typical of troodontids, forming the ventral margin of the antorbital fenestra. The maxillary ventral ramus lacks its ventral portion, and is shattered with only its posterior portion preserved as two shelves (Fig. 4B). The two shelves seemingly form a groove to receive the anterior end of the suborbital process of the jugal, as reported in Liaoningvenator (Shen et al., 2017b). Medially, the palatal shelf is well developed with a vaulted medial margin (Fig. 4B), and possibly contacts the maxillary process of the palatine. A foramen pierces through the middle portion of the palatal shelf of the maxilla (Fig. 4B).

Nasal

Only the posterior portions of the nasals are preserved (Fig. 4A). The maximum transverse width of the nasals is 7.9 mm. The dorsal surface of the nasal is smooth. As in Almas (Pei et al., 2017a), Byronosaurus (Makovicky et al., 2003) and Saurornithoides (Norell et al., 2009), a row of foramina develops on the anterior part of the dorsal surface of the nasal (Fig. 4A), and opens into the nasal cavity. As in the holotype (Xu et al., 2002) and Sinusonasus (Xu & Wang, 2004), the anterolateral edge of the nasal expands laterally above the antorbital fenestra, forming a small lateral shelf that overlaps the maxilla and the lacrimal (Fig. 3). The nasal is slightly vaulted medial to the lateral shelf. A ridge participates to the lateral wall of the lacrimal duct ventral to the nasal lateral shelf. Posterior to the shelf, the nasal articulates with the lacrimal along a slightly sigmoidal suture in dorsal view (Fig. 4A). The posterior end of the nasal reaches the level of the preorbital bar. As in Zanabazar (Norell et al., 2009) and Liaoningvenator (Shen et al., 2017b), the posterior parts of the nasals seemingly form a V-shaped notch in dorsal view (Fig. 4A), overlapping the frontals.

Lacrimal

The lacrimal is well preserved on the right side (Figs. 3 and 5). As in other deinonychosaurians, this bone is “T”-shaped with an anterior process, a posterior process and a preorbital bar (ventral process). The anterior and posterior processes are dorsally positioned along the skull roof. Medially, a large fossa is present at the junction of the anterior process, the posterior process and the preorbital bar (Fig. 5B).

Figure 5 CT-rendered left lacrimal of PMOL-AD00102 in lateral (A) and medial (B) views.

Study sites: fo, fossa; lap, anterior process of lacrimal; lf, lacrimal foramen; ld, lacrimal duct; lpp, posterior process of lacrimal; pb, preorbital bar; soc, supraorbital crest.

As in other troodontids (Turner, Makovicky & Norell, 2012), the anterior process is longer than the posterior process (Fig. 5), though the exposed portion of the anterior process is almost as long as the posterior process because the anterior tip of the anterior process is obscured by the nasal anterolateral shelf in dorsal and lateral views (Figs. 3 and 4A). As in Jianianhualong (Xu et al., 2017), Sinusonasus (Xu & Wang, 2004) and Almas (Pei et al., 2017a), the anterior process is similar in length to the preorbital bar (Fig. 5). The anterior process points anteroventrally and makes an acute angle with the preorbital bar in lateral view. The anterior process has limited contact with the maxilla rostrally, and forms most of the dorsal margin of the antorbital fenestra, as in Xixiasaurus (Lü et al., 2010) and Byronosaurus (Makovicky et al., 2003). The lacrimal duct is developed along the anterior process lateroventrally, and the duct opens laterally on the junction between the anterior process and the preorbital bar (Fig. 5), as in Mei (Gao et al., 2012), Byronosaurus (Makovicky et al., 2003), Troodon (Currie, 1985), and Sinornithoides (Currie & Dong, 2001), but in contrast to dromaeosaurids and other non-avian theropods in which the lacrimal duct penetrates the preorbital bar (Currie & Dong, 2001; Pei et al., 2014). Dorsal to the lacrimal foramen, the anterior process has a lateral extension (Fig. 5A), as in Mei (Gao et al., 2012). Ventral to the lacrimal foramen, a small shallow depression is present (Fig. 5A).

The posterior process is mediolaterally broad, forming the anterodorsal border of the orbit. The posterior process projects posterodorsally, making an obtuse angle with the preorbital bar. It bears a laterally expanded supraorbital crest anterodorsal to the orbit (Figs. 3, 4A and 5A), as in most troodontids (Pei et al., 2017a). The dorsal surface of the posterior process is smooth, in contrast to the rugose condition in Dromaeosaurus (Currie, 1995). The posterior process is bifurcated with a longer dorsal ramus in lateral view (Figs. 3 and 5A) as in Jianianhualong (Xu et al., 2017). The medial surface of the posterior process bears a shallow and sub-triangular groove that widens posteriorly between the dorsal and ventral rami (Fig. 5B).

The preorbital bar forms the posterior margin of the antorbital fenestra, and slightly curves anteroventrally at its ventral portion (Fig. 2). As in dromaeosaurids, the preorbital bar does not contact the maxilla ventrally (Currie, 1995). The preorbital bar is everted, and the lateral surface of its upper portion becomes the posterior surface at the lower portion. The lower portion of the preorbital bar becomes anteroposteriorly compressed. A distinct groove extends ventrally along the posterolateral surface of the preorbital bar. Anterior to the groove, a lateral flange is present along the anterolateral surface of the preorbital bar (Figs. 2, 3 and 5A), as in other troodontids (Xu et al., 2017). The ventral end of the preorbital bar locates in a long and shallow groove on the jugal, which makes the preorbital bar seemingly able to slide along this groove. The preorbital bar makes a right angle with the suborbital process of the jugal (Fig. 2).

Postorbital

The left postorbital is incompletely preserved and its posterior process is missing (Fig. 2). The anterior process of the postorbital is fragmentary, and probably upturns and contacts the postorbital process of the frontal, based on the upturned articular surface of the postorbital process of the frontal. Laterally, the main body of the postorbital is depressed. The anterior edge of the postorbital curves and forms the posterodorsal margin of the orbit. The distal part of the ventral process is missing but possibly articulates with the postorbital process of the jugal.

Squamosal

The left squamosal is well preserved, only missing its rostral process, and the right squamosal is represented by a medial process (Figs. 2 and 3). The main body of the squamosal wraps the quadrate head with an articular cotylus, and bears a distinct lateral recess as in derived troodontids such as Almas (Pei et al., 2017a) and Linhevenator (Xu et al., 2011). The quadratojugal process of the squamosal tapers ventrally in lateral view. The anterior edge of the quadratojugal process is mediolaterally thinner than its posterior edge as in Troodon (Currie, 1985). Distally, this process is isolated from the quadrate shaft, likely due to taphonomic distortion. However, it possibly would have contacted the upper portion of the quadrate shaft in life. The preserved posterior process of the squamosal is downturned and wedged between the quadrate anteriorly and the paroccipital process posteriorly. The medial process of the squamosal articulates with the anterior surface of the nuchal crest formed by the parietal.

Jugal

The left jugal is well preserved (Figs. 2, 6A and 6B). The jugal of the new specimen is triradiate, with a suborbital process, a postorbital process and a quadratojugal process as in other deinonychosaurians, e.g., Gobivenator (Tsuihiji et al., 2014), Almas (Pei et al., 2017a), Microraptor (Pei et al., 2014) and Velociraptor (Barsbold & Osmólska, 1999). The anteroposterior length of the left jugal is 56.8 mm.

Figure 6 CT-rendered left jugal of PMOL-AD00102 in dorsal (A) and medial (B) views, and a cross-sectional CT image of jugal (C).

Study sites: dp, dorsal prong of quadratojugal process of jugal; ect, ectopterygoid contact; jd, depression on jugal; jf, fossa on jugal; jg, groove on jugal; jt, trough on jugal; ms, medial shelf on jugal; pop, postorbital process of jugal; qjp, quadratojugal process of jugal; sop, suborbital process of jugal; vp, ventral prong of quadratojugal process of jugal.

The anterior tip of the suborbital process inserts into the ventral ramus of the maxilla (Fig. 2). In lateral view, the suborbital process tapers anteriorly, and contributes to the posteroventral corner of the antorbital fenestra. In dorsal view, the dorsal margin of the suborbital process is slightly convex laterally (Fig. 6A). The suborbital process is dorsoventrally shallow ventral to the antorbital fenestra and the anterior half of the orbit. It becomes dorsoventrally deep ventral to the posterior half of the orbit, reaching twice the depth of its anterior portion (Fig. 2). Posterior to the antorbital fenestra, the suborbital process bears a longitudinal ridge along its ventral portion of the lateral surface which terminates below the midpoint of the orbit as in Linhevenator (Xu et al., 2011). A longitudinal groove is developed in the lateral surface of the suborbital process dorsal to this ridge as in the holotype (Xu, 2002), Mei (Xu & Norell, 2004), Linhevenator (Xu et al., 2011), and Zanabazar (Norell et al., 2009). Ventral to this ridge, a shallow and narrow groove is developed on the lateroventral surface of the suborbital process (Fig. 4B), and this groove starts below the preorbital bar and terminates posteriorly below the midpoint of the suborbital portion of the suborbital process. The suborbital process has a medial shelf (Figs. 4B and 6) close to the ventral margin extending from the point just anterior to the preorbital bar to the level anterior to the expanded suborbital portion of the jugal, and this feature is reported in troodontids for the first time. A shallow groove is developed dorsal to the shelf (Fig. 6A). This groove articulates with the preorbital bar of the lacrimal. Medially, a deep fossa is present at the posterior end of the groove and dorsal to a depression (Figs. 6A and 6B). A shallow trough is developed anteroventral to the depression, separated from the groove by the medial shelf (Figs. 6A and 6B). Further anteriorly, a rough articular surface for the ectopterygoid is located on the medial surface of the dorsoventrally thickened portion of the medial shelf (Fig. 6B).

The postorbital process slightly inclines posterodorsally, and the dorsal half of the postorbital process is fragmentary (Fig. 6B). The postorbital process has a broad and anterolaterally oblique surface possibly for articulating with the jugal process of the postorbital (Fig. 6A). In medial view, a low ridge develops on the postorbital process of the jugal as in the holotype (Xu, 2002). This ridge terminates at the base of the postorbital process (Figs. 6A and 6B).

The quadratojugal process tapers posteriorly and splits into two prongs for the reception of the jugal process of the quadratojugal (Figs. 2 and 6B). The dorsal prong is longer than the ventral prong. The jugal process of the quadratojugal articulates with the lateral surface of the dorsal prong and the medial surface of the ventral prong (Fig. 6B).

Quadratojugal

The left quadratojugal is preserved, and it is comprised of a jugal process and a squamosal process (Fig. 2). In lateral view, the quadratojugal is reversed L-shaped as in Sinornithoides (Russell & Dong, 1993), Almas (Pei et al., 2017a), Archaeopteryx (Elzanowski & Wellnhofer, 1996) and Anchiornis (Pei et al., 2017b), different from the inverted “T”-shaped quadratojugal in dromaeosaurids (Currie, 1995). Unlike dromaeosaurids, the quadratojugal does not contact the squamosal (Fig. 2), in agreement with Mei (Xu & Norell, 2004), Sinornithoides (Russell & Dong, 1993) and Gobivenator (Tsuihiji et al., 2014). The main body of the quadratojugal covers the ventral portion of the quadrate laterally and bears a socket on its anterodorsal surface. The jugal process of the quadratojugal is damaged, with only the anterior-most portion that inserts into a slot on the quadratojugal process of the jugal preserved. The squamosal process is slender and dorsally projected. As in Sinornithoides (Russell & Dong, 1993), the squamosal process wraps the quadrate shaft posteriorly.

Quadrate

The left quadrate is slightly fractured and the right quadrate lacks the anterior part of its pterygoid ramus (Figs. 4B and 7). The quadrate has a height of approximately 20 mm.

The quadrate head is singular in dorsal view. It is anteromedial-posterolaterally wide, and is wrapped by the squamosal. The quadrate head is exposed extensively in lateral view (Fig. 2). Unlike dromaeosaurids, the quadrate body does not have a triangular lateral process. The anterior surface of the quadrate body above the mandibular articulation is concave in anterior view. In posterior view, the pneumatic fenestra is located in the middle portion of the quadrate body (Figs. 7A and 7C) as in the holotype (Xu et al., 2002) and other troodontids (Makovicky & Norell, 2004). The quadrate shaft bears a strong posterior curvature. In lateral view, the dorsal half of the quadrate shaft is wider than its ventral half and has a smooth lateral surface. The quadrate ridge is developed medially along the quadrate shaft (Fig. 7C).

Figure 7 CT-rendered palatal elements of PMOL-AD00102.

(A) palate in dorsal view; (B) left palatal elements in lateral view; (C) left palatal elements in medial view. Study sites: e, epipterygoid; ec, ectopterygoid; in, internal naris; iptv, interpterygoid vacuity; pf, palatine fenestra; pl, palatine; ppf, posterior pneumatic fenestra; pt, pterygoid; ptf, pterygopalatine fenestra; q, quadrate; qr, quadrate ridge; stf, subtemporal fenestra; v, vomer.

Two asymmetric condyles are present for the mandibular articulation. A shallow diagonal sulcus separates these two condyles (Fig. 4B). The medial condyle is larger than the lateral condyle, similar to the condition in the unnamed Early Cretaceous troodontid IGM 100/44 (Barsbold, Osmólska & Kurzanov, 1987), Saurornithoides (Norell & Hwang, 2004), Dromaeosaurus (Colbert & Russell, 1969) but unlike the condition in Sinornithosaurus (Xu & Wu, 2001) and Velociraptor (Barsbold & Osmólska, 1999) in which the lateral condyle is larger. Dorsolateral to the lateral condyle, the quadrate bears a sub-trapezoidal facet that is overlapped by the quadratojugal.

Laterally, the pterygoid ramus is sheet-like and overlaps the quadrate process of the pterygoid. The dorsal edge of the pterygoid ramus descends anteriorly and is thickened as in Sinornithosaurus (Xu, 2002). In medial view, the pterygoid ramus bears a concavity that becomes larger and wider ventrally. Anterior to the concavity, an anteriorly bowed low ridge defines the posterior boundary of the articular surface with the quadrate process of the pterygoid (Fig. 7C).

Pterygoid

The left pterygoid is nearly completely preserved, and the right pterygoid is missing its main body and the anterior portion of the quadrate ramus (Fig. 7). Dorsally, the anterior end of the pterygoid seemingly contacts the other pterygoid, and a long and tear-shaped interpterygoid vacuity is present along the midline of the palate (Fig. 7A). It is unclear whether the two pterygoids contact posteriorly due to the incompleteness of the right pterygoid. However, such contact is unlikely based on the shape of the left pterygoid. If this morphology is correctly interpreted here, it would resembles the condition in Archaeopteryx (Mayr et al., 2007) and some dromaeosaurids, such as Deinonychus (see Fig. 5 in Ostrom, 1969) and Dromaeosaurus (see Fig. 1C in Currie, 1995), but unlike the condition in Saurornithoides (Norell et al., 2009) and Gobivenator (see Fig. 5 in Tsuihiji et al., 2014), in which the two elements contact and nearly contact with each other, respectively. The anterior (palatine) ramus is vertical and long, forming the medial margin of the pterygopalatine fenestra. The anterior half of the anterior ramus deepens anteriorly, while its posterior half becomes a slender rod (Fig. 7C). In lateral view, the anterior ramus bears a narrow and shallow trough along the posterior half of the ramus. Posterior to the anterior ramus, the main body of the pterygoid expands laterally and becomes a thin sheet (Fig. 4B). The pterygoid flange develops as a distinct lateral process at the posterior end of the main body, as in Gobivenator (Tsuihiji et al., 2014) and Saurornithoides (Norell et al., 2009), but different from the posteriorly curved flange in Almas (Pei et al., 2017a). A prominent projection develops ventral to the pterygoid flange, as in the holotype (Xu, 2002). This projection is shorter than the pterygoid flange. Posteriorly, the pterygoid has an articular facet for the basipterygoid process of the basisphenoid, formed by a short medial process and the quadrate ramus.

The quadrate ramus is shelf-like and bifurcates in medial view (Fig. 7C), as in Sinornithosaurus (Xu & Wu, 2001; Xu, 2002) and Archaeopteryx (Elzanowski & Wellnhofer, 1996). The dorsal process is longer than the ventral process, and the dorsal margin of the dorsal process is thickened. The quadrate ramus contacts the pterygoid ramus of the quadrate laterally. The lateral surface of the quadrate ramus bears an oblique ridge and its medial surface is concave.

Vomer

Only the paired pterygoid rami of the vomers are preserved (Fig. 7), which extend posteriorly to the level of the last dentary tooth, and therefore it is impossible to determine the degree of the fusion of the vomers. The pterygoid ramus is a vertical plate as in Dromaeosaurus (Currie, 1995) and Archaeopteryx (Elzanowski & Wellnhofer, 1996). The contact between the vomer and the pterygoid is seemingly akinetic because the suture between the two bones is hardly discernible, unlike the less tightly joined contact in Velociraptor (Barsbold & Osmólska, 1999).

Palatine

The palatines are well preserved (Figs. 2, 3 and 7). As in other non-avian theropods, the palatine is tetraradiate and comprised of a vomeropterygoid process, a maxillary process, a jugal process and a pterygoid process. As in dromaeosaurids (Norell & Makovicky, 2004), Gobivenator (see Fig. 5 in Tsuihiji et al., 2014) and Archaeopteryx (Elzanowski, 2001), the palatine forms the lateral margin of the long pterygopalatine fenestra (Fig. 7). As in Archaeopteryx (Mayr et al., 2007; Rauhut, Foth & Tischlinger, 2018), an anterior triangular depression and a posterior sub-triangular depression are formed on the main body of the palatine and are separated by a prominent transverse crest that reaches the base of the jugal process (Fig. 3). This is distinguished from that in Velociraptor (Barsbold & Osmólska, 1999), Deinonychus (Ostrom, 1969), and Gobivenator (Tsuihiji et al., 2014), in which the transverse crest is absent. A canal opens into the maxillary process at the anterior end of the anterior depression. Another canal passes into the transverse crest at the anterodorsal end of the posterior depression, and terminates at the posterodorsal end of the anterior depression.

In lateral view, the maxillary process is long and slender with an anterior upturning end, forming the posterior and the lateral margins of the internal naris. This process is longer than the vomeropterygoid process as in Gobivenator (Tsuihiji et al., 2014) and Archaeopteryx (Mayr et al., 2007), but unlike the condition in Deinonychus (Ostrom, 1969) and Velociraptor (Barsbold & Osmólska, 1999), in which the process is shorter. A shallow lateroventral trough is developed on the maxillary process, possibly for contacting the maxilla. This trough widens posteriorly, and is dorsally and posteriorly defined by a laterally directed lamina.

The vomeropterygoid process is dorsoventrally deep and anteriorly hooked, and it is vertically oriented (Figs. 7B and 7C), constituting the medial edge of the internal naris (Fig. 7A). A prominent ridge of the vomeropterygoid process develops dorsally and twists posteromedially. The medial surface of the vomeropterygoid process is smooth, possibly for contacting with the other palatine. A small vertical shelf is present posterior to the smooth medial surface, and bears a sub-triangular depression. Ventral to this vertical shelf, a large groove is present along the vomeropterygoid process and reaches the base of the pterygoid process.

The jugal process is short and sub-triangular, forming the anterolateral margin of the palatine fenestra, just as in Gobivenator (Tsuihiji et al., 2014), Deinonychus (Ostrom, 1969), Velociraptor (Barsbold & Osmólska, 1999) and Archaeopteryx (Mayr et al., 2007). Posteriorly, the jugal process contacts the jugal. The ventral surface of the jugal process is smooth.

The pterygoid process is twice as long as the vomeropterygoid process, extending posteriorly for contacting with the pterygoid and the ectopterygoid, contributing to the medial margin of the palatine fenestra (Fig. 7A). The anterior half of the pterygoid process is band-like with a curved lateral border, but the posterior half of the pterygoid process widens posteriorly.

Ectopterygoid

The left ectopterygoid is preserved (Fig. 7). It consists of a jugal process, an ectopterygoid (pterygoid) flange and a pterygoid process. The jugal process is hooked and contacts the medial surface of the jugal below the orbital margin, as in other non-avian theropods. This process sharpens posteriorly, and separates the palatine fenestra from the subtemporal fossa (Ostrom, 1969). The jugal process extends posteriorly almost to the level of the posterior end of the pterygoid process, in contrast to Linhevenator (Xu et al., 2011) and Archaeopteryx (Elzanowski & Wellnhofer, 1996) in which the process is distinctly shorter than the pterygoid process. The medial portion of the jugal process is short, and therefore the space between the jugal process and the pterygoid process is mediolaterally narrow, unlike Linhevenator (Xu et al., 2011), Jianianhualong (see Fig. 2 in Xu et al., 2017) and Archaeopteryx (Elzanowski & Wellnhofer, 1996) in which this space is large. The pterygoid flange is robust and extends posteroventrally. A groove is present on the pterygoid flange in lateral view. Medial to the pterygoid flange, a deep pocket excavates the ventral surface of the pterygoid process, as in other non-avian theropods. The pterygoid process is horizontally oriented, overlapped by the pterygoid process of the palatine, as in Archaeopteryx (Elzanowski & Wellnhofer, 1996). The pterygoid process overlaps the main body of the pterygoid immediately anterior to the quadrate ramus of the pterygoid. The pterygoid process is wider than long as in Archaeopteryx (Elzanowski & Wellnhofer, 1996). Dorsally, a depression occupies most of the dorsal surface of the pterygoid process, and a ridge separates a narrow and deep groove from the depression posteriorly, as in Linhevenator (Xu et al., 2011), but unlike the condition in Dromaeosaurus (Currie, 1995), Velociraptor (Barsbold & Osmólska, 1999), and Tsaagan (Norell et al., 2006) in which such a depression is absent. This condition is also different from Deinonychus (Ostrom, 1969) and Saurornitholestes (Sues, 1978) in which a pit and two little depressions are present respectively. Interestingly, the dorsal depression seems connecting the ventral pocket through some foramina like in Saurornitholestes (Sues, 1978), though it is difficult to known whether this condition is a preservational artifact in PMOL-AD00102.

Epipterygoid

The left epipterygoid is preserved (Figs. 2 and 7). This is the first report of an epipterygoid in troodontids and the shape of the bone is similar to that in Archaeopteryx (Rauhut, 2014) and other non-avian theropods. This bone is laterally visible through the orbit. The epipterygoid bears a shallow fossa on the medial surface that is anterodorsally bordered by a thickened ridge (Fig. 7C). Medially, the epipterygoid overlaps the anterodorsal surface of the quadrate ramus of the pterygoid, unlike Dromaeosaurus in which the epipterygoid overlaps the dorsal rim of the quadrate ramus of the pterygoid (Colbert & Russell, 1969). Dorsally, the epipterygoid has a pointed laterosphenoid process, and this process possibly contacts the laterosphenoid, inferred from a depression on the laterosphenoid.

Frontal

The left and right frontals are well-preserved. The posterolateral portion of the right frontal is partially fractured (Figs. 3 and 4A). The anteroposterior length of the frontal is 34.4 mm, approximately three times of the minimum width between the orbits. This ratio is similar to that of Mei, but is larger than that in Jianianhualong (1.8 times; Xu et al., 2017).

The frontal is subtriangular in dorsal view and forms the dorsal margin of a large and circular orbit. The lateral margin of the frontal sharpens anteriorly (Fig. 8A). A slot is present on the anterolateral end of the frontal (Fig. 8) as seen in dromaeosaurids (Xu & Wu, 2001). However, this slot is not for the lacrimal attachment in this specimen and only defined by a anterolateral prong, different from the condition in dromaeosaurids, in which the posterior process of the lacrimal is attached onto this slot (Currie, 1995). This anterolateral prong of the frontal possibly represents the vertical lamina in the holotype (Xu et al., 2002), and this variation is probably preservational. The orbital margin of the frontal is vaulted and rugose, as in other troodontids (Currie, 1985; Norell et al., 2009; Lü et al., 2010; Tsuihiji et al., 2014; Pei et al., 2017a) and most dromaeosaurids (Currie, 1995; Norell et al., 2006; Xu et al., 2015a). Dorsally, a shallow trough is developed lateral to the suture between the frontals, and a longitudinal ridge is present along the midline of each frontal as in Zanabazar (Norell et al., 2009). A distinct postorbital process diverges gently from the orbital rim (Fig. 4A), different from the sharp emargination in dromaeosaurids (Currie, 1987a). The distal end of the postorbital process is broadly notched between an anterior projection and a posterior projection (Fig. 4A). A depression is present on the dorsal surface of the postorbital process, and possibly medially continuous with the supratemporal fossa, like in Zanabazar (Norell et al., 2009), Troodon (Currie, 1985), Linhevenator (Xu et al., 2011) and some dromaeosaurids (Barsbold & Osmólska, 1999; Xu & Wu, 2001). The anterior margin of the supratemporal fossa is straight, defined by a transverse ridge on the frontal that reaches onto the postorbital process (Fig. 4A), as in other troodontids but in contrast with a sigmoidal boundary in dromaeosaurids (Norell & Makovicky, 2004). Posterior to this ridge, the frontal slopes down gently, different from a steep slope in Troodon (Currie, 1985) and Zanabazar (Norell et al., 2009). The frontal-parietal suture is sigmoidal.

Figure 8 CT-rendered braincase of PMOL-AD00102 in right lateral view (A) and ventral view (B).

Study sites: bpt, basipterygoid process; bptr, basipterygoid recess; br, basisphenoid recess; bt, basal tuber; cc, crista cranii; cp, cultriform process; dr, dorsal tympanic recess; f, frontal; fo, fossa; ls, laterosphenoid; nc, nuchal crest; oc, occipital condyle; p, parietal; pop, paroccipital process; pro, prootic; psr, parasphenoid recess; sc, sagittal crest; scr, subcondylar recess; sf, slot on frontal; sor, subotic recess.

The crista cranii is well developed and observable in lateral and ventral views (Fig. 8). The posterior portion of the crista cranii is deep and inclines medioventrally, whereas the anterior portion is shallow and vertical. The crista cranii forms the lateral wall of the trough for the olfactory tract and olfactory bulb. This trough is shallow at the anteriormost part, and becomes deeper and wider posteriorly. A shallow shelf separates the left and right troughs along the midline of the frontals (Fig. 8B).

Parietal

The parietals are fused as in other troodontids (Fig. 4A). The parietal is fused with the supraoccipital posteriorly. The sagittal crest is high and lamina-like along the midline of the parietals (Fig. 8A), as in Zanabazar and Troodon (Norell et al., 2009), in contrast to the condition in the holotype (Xu et al., 2002), Jianianhualong (Xu et al., 2017), Liaoningvenator (Shen et al., 2017b) and Linhevenator (Xu et al., 2011) where the crest is low, and distinct from Mei (Xu & Norell, 2004) in which such a crest is absent. The dorsal surface of the parietal has a gentle slope lateral to the sagittal crest. In lateral view, the suture between the parietal and the laterosphenoid is roughly straight like that in the holotype (Xu, 2002). The nuchal crest (Fig. 8A) is well developed with a similar depth of the sagittal crest, but it is distinctly longer than the sagittal crest. The nuchal crest is slightly sigmoidal in dorsal view (Fig. 4A), defining the posterior boundary of the supratemporal fossa and the dorsal margin of the occiput.

Braincase

The occiput is well-preserved. The bones forming the occiput are fused, and the sutures between the supraoccipital, the exoccipital and the basioccipital are not identifiable. The occiput inclines slightly anteriorly as preserved in this specimen. The foramen magnum is larger than the occipital condyle and dorsoventrally higher than wide (Fig. 9) as in the holotype (Xu et al., 2002), other troodontids (Xu, 2002) and some dromaeosaurids (e.g., Tsaagan; Norell et al., 2006). Dorsal to the foramen magnum, two foramina probably represent the openings for cerebral veins (Fig. 9). The occipital condyle has a constricted neck (Fig. 8B) as in Troodon and Zanabazar (Norell et al., 2009). Two foramina represent the openings of the CN XII immediately lateral to the occipital condyle (Fig. 9). Lateral to the openings of CN XII, a larger foramen deriving from the metotic strut represents the exit of CN X and CN XI (Fig. 9).

Figure 9 CT-rendered braincase of PMOL-AD00102 in posterior view.

Study sites: bpt, basipterygoid process; bt, basal tuber; f, frontal; fm, foramen magnum; ls, laterosphenoid; nc, nuchal crest; p, parietal; pop, paroccipital process; oc, occipital condyle; vcmp, posterior canal of middle cerebral vein; X, XI, tenth and eleventh cranial nerve exit; XII, twelfth cranial nerve exit.

The exoccipital is fused with the opisthotic. The paroccipital process is short and distally pendulous, as in Mei (Xu & Norell, 2004), and extends lateroventrally. The distal end of the paroccipital process is ventral to the level of the ventral margin of the occipital condyle. In contrast, the paroccipital process is long, straight and extends laterally or posterolaterally in dromaeosaurids (Turner, Makovicky & Norell, 2012). The base of the paroccipital process is constricted (Fig. 9) as in the holotype (Xu, 2002).

The basioccipital is complete. It is co-ossified with the basisphenoid-parasphenoid anteriorly. Anterior to the occipital condyle, a subcondylar recess is developed in the basioccipital (Fig. 8B) as in Anchiornis (Pei et al., 2017b). More anteriorly, two reduced basal tubera are present, but seem confluent with each other by a septum (Fig. 9), unlike the holotype and other troodontids in which a V-shaped notch is present between the two tubera (Xu, 2002). The posterior surface of the basal tubera is concave, which is probably a homologous structure to the V-shaped notch. This variation between PMOL-AD00102 and the holotype could be preservation or ontogeny related.

The basisphenoid is co-ossified with the parasphenoid anteriorly, and contacts the pterygoid by two diverging basipterygoid processes. Unlike other troodontids, but similar to dromaeosaurids (Norell & Makovicky, 2004), Anchiornis (Pei et al., 2017b) and Archaeopteryx (Rauhut, 2014), a basisphenoid recess is developed (Fig. 8B). Two fossae are present lateral to the posterior end of the basisphenoid recess (Fig. 8B). This represents a novel character that has not been reported in other troodontids. As in the holotype (Xu et al., 2002) and Liaoningvenator (Shen et al., 2017b), the basipterygoid process is solid and the basipterygoid recess is well developed on the dorsolateral surface of the basipterygoid process (Figs. 8 and 10). In contrast, the basipterygoid process in the Late Cretaceous troodontids is hollow, and the basipterygoid recess is absent (Turner, Makovicky & Norell, 2012). The distal end of the basipterygoid process is blunt, unlike the pointed end in the holotype (Xu et al., 2002). As in the holotype (Xu, 2002), the basipterygoid process directs lateroventrally, but unlike the condition in Troodon, in which the process is relatively posteriorly directed (Currie & Zhao, 1993).

The anterior end of the parasphenoid is posterior to the lacrimal as preserved in the specimen. As in Velociraptor (Barsbold & Osmólska, 1999), the preserved anterior portion of the cultriform process is V-shaped in cross section. The base of the parasphenoid is not bulbous, as in the holotype (Xu et al., 2002), and the pituitary fossa is well preserved. Laterally, unlike the holotype (Xu, 2002), the otosphenoidal crest (Fig. 10) is developed and defines a “lateral depression” (parasphenoid recess; Xu, 2002) as in other troodontids (Makovicky & Norell, 2004). As in Byronosaurus (Makovicky et al., 2003), the lateral depression is bordered posteriorly by the subotic recess. The parasphenoid recesses on both sides are large and highly pneumatized, and connect with each other medially. As in the holotype (Xu, 2002), the parasphenoid recess is divided into two openings by an ossified bar, for accommodating the pituitary fossa and the internal carotids (Fig. 10). The anterior opening is oval, and the posterior one (the anterior tympanic recess) is dorsoventrally elongated. Ventral to the divergence of the two ossified carotid canals, a bar extends ventrally into the basisphenoid and this bar possibly represents a neomorph (Figs. 8A and 10).

Figure 10 CT-rendered braincase of PMOL-AD00102 in left lateral view.

Study sites: at, accessory tympanic recess; bpt, basipterygoid process; bptr, basipterygoid recess; cif, crista interfenestralis; ctr, caudal tympanic recess; dr, dorsal tympanic recess; fo, fenestra ovalis; fpr, fenestra pseudorotunda; ls, laterosphenoid; mf, metotic fissure; nc, nuchal crest; oc, occipital condyle; otc, otosphenoidal crest; p, parietal; pld, perilymphatic duct; pls, pit on laterosphenoid; pop, paroccipital process; pro, prootic; psr, parasphenoid recess; III, third cranial nerve exit; IV, fourth cranial nerve exit; V, fifth cranial nerve exit; VII, seventh cranial nerve exit.

The laterosphenoid is a relatively large bone forming the anterolateral wall of the braincase. The laterosphenoid is co-ossified with the orbitosphenoid and the basisphenoid-parasphenoid ventrally. As in Troodon (Currie, 1985), the postorbital process of the laterosphenoid contacts the frontal dorsally and has a smooth distal capitulum for contacting the postorbital. The laterosphenoid forms the lateral wall of the braincase and is almost vertical and smooth. A pit develops on the ventral surface of the laterosphenoid (Fig. 10) ventrally as in other troodontids (Makovicky et al., 2003). Medial to the pit, two foramina are present, representing the exits of CN IV and CN III respectively (Fig. 10). The exit for CN IV is higher than that of CN III.

Posterior to the laterosphenoid, the prootic forms the posterolateral wall of the braincase. The prootic is co-ossified with the basisphenoid ventrally and the opisthotic posteriorly. There is a ridge defining the anterior margin of the dorsal tympanic recess on the suture between the prootic and the laterosphenoid. As in the holotype (Xu et al., 2002), the dorsal tympanic recess is a large and shallow depression (Figs. 8A and 10). Anteroventral to the dorsal tympanic recess, two openings are separated from each other by a mound as in Troodon (Norell, Makovicky & Clark, 2000). The anterior one of these two openings represents the exit of CN V and the posterior one represents the exit of CN VII (Fig. 10). As in Byronosaurus and Almas (Norell, Makovicky & Clark, 2000; Pei et al., 2017b), CN VII and the middle ear cavity are located dorsal to the rim of the lateral depression (Fig. 10), but in contrast to the condition in Zanabazar and Saurornithoides (Norell et al., 2009), in which CN VII and the middle ear cavity are located within the lateral depression. As in the holotype (Xu, 2002) and Troodon (Turner, Makovicky & Norell, 2012), the fenestra ovalis and the fenestra pseudorotunda are separated from each other by the crista interfenestralis which is depressed within the middle ear cavity (Fig. 10), different from Byronosaurus in which the crista interfenestralis is flush with the lateral surface of the prootic (Makovicky et al., 2003). Posterior to the fenestra pseudorotunda, the metotic fissure penetrates the lateral wall of the braincase (Fig. 10) as in Troodon (Currie & Zhao, 1993), Byronosaurus (Makovicky et al., 2003), and the unnamed troodontid IGM 100/44 (Barsbold, Osmólska & Kurzanov, 1987). As in Byronosaurus (Makovicky et al., 2003), the hypoglossal nerve possibly emerges from the braincase through this fissure. As in Byronosaurus (Makovicky et al., 2003), a small foramen is developed on the anterior wall of the metotic fissure at the midheight (Fig. 10) and is possibly the opening of the perilymphatic duct (Makovicky & Norell, 1998). Dorsal to the metotic fissure, the accessory tympanic recess is present (Fig. 10). As in the holotype (Xu, 2002), the caudal tympanic recess is seemingly confluent with the accessory tympanic recess through a shallow groove.

The inner surface of the braincase is reconstructed by the CT-scan images, though the sutures are undetectable as the bones forming the braincase are co-ossified as mentioned above. On the inner surface of the braincase, the laterosphenoid shows a large and well-developed fossa for accommodating the optic lobe (Fig. 11A). Posterior to the fossa, a groove represents the passage for the middle cerebral vein that emerges posteriorly from the braincase through its posterior canal (Fig. 11A). Ventral to the groove, the floccular recess is large and deep (Figs. 11A and 11B). An inner opening for CN VII is present ventral to the floccular recess. A large opening represents the exit of CN V anterior to CN VII (Fig. 11B). Posteroventral to the floccular recess, two foramina are developed on the medial wall of the inner ear (Fig. 11B). The dorsal foramen is smaller, possibly represents the endolymphatic duct. The ventral foramen is larger and is divided into two foramina entering the inner ear as the opening for CN VIII. The upper foramen and the lower foramen are for the vestibule branch and cochlear branch of CN VIII respectively. Posterior to the CN VIII, a small foramen is present as the perilymphatic duct (Fig. 11B). Further posteriorly, as in Byronosaurus (Makovicky et al., 2003), the metotic fissure shows an hourglass shape and has a constriction slightly below its midheight (Fig. 11B). Further posterior to the metotic fissure, two openings for CN XII are present and the upper one is larger than the lower one (Fig. 11B).

Figure 11 CT-rendered braincase of PMOL-AD00102 in medial view.

(A) anterior part of the right side; (B) posterior part of the left side. Study sites: ed, endolymphatic duct; fopt, fossa of optic lobe; fr, floccular recess; mf, metotic fissure; oc, occipital condyle; pld, perilymphatic duct; vcm, groove for middle cerebral vein; vcmp, posterior canal of middle cerebral vein; V, fifth cranial nerve exit; VII, seventh cranial nerve exit; VIII, eighth cranial nerve exit; XII, twelfth cranial nerve exit.

Stapes

The left stapes is preserved, represented by a proximal shaft and a footplate (Fig. 12). The stapes is reported in troodontids for the first time. As in tyrannosaurids (Witmer & Ridgely, 2009) and oviraptorids (Clark, Norell & Rowe, 2002), no groove is present in the paroccipital process to receive the stapes in PMOL-AD00102, but in contrast to the condition in dromaeosaurids (Colbert & Ostrom, 1958; Currie, 1995) in which such a groove is present. The stapes is a slender bone with a small footplate fitting the shape of the fenestra ovalis. The stapes projects both posterolaterally and ventrally, as in tyrannosaurids (Witmer & Ridgely, 2009) and oviraptorids (Clark, Norell & Rowe, 2002) but unlike the posterolaterally directed condition in dromaeosaurids (Colbert & Ostrom, 1958).

Figure 12 CT-rendered left stapes of PMOL-AD00102 in lateral (A) and dorsal (B) views.

Study site: ft, footplate.

The footplate is subtriangular (Fig. 12A), and its maximum diameter is about 1.7 mm. The maximum diameter of the footplate is about four times the diameter of the shaft (Fig. 12A). The shaft of the stapes is cylindrical, and it attaches laterally on the footplate, at a position slightly anteroventral to the midpoint of the footplate (Fig. 12A). The shaft is almost vertical to the footplate, and is only slightly posteriorly oblique (Fig. 12B). A small bar medial to the medial condyle of the left quadrate is recognized as the distal portion of the stapes, as inferred from its slender shape (Fig. 4B). If this interpretation is correct, the stapes has a tapering distal end.

Mandible

Both mandibular rami are well-preserved at the middle-posterior portion (Fig. 13). The dorsal margin of the post-dentary portion of the mandibular ramus is nearly straight in lateral view, while the ventral margin is slightly bowed. The external mandibular fenestra is large and elongated with a length of 28 mm (Fig. 13A), as in Saurornithoides (Osborn, 1924), Sinornithoides (Russell & Dong, 1993), Gobivenator (Tsuihiji et al., 2014), Velociraptor (Barsbold & Osmólska, 1999), Microraptor (Pei et al., 2014) and Tsaagan (Turner, Makovicky & Norell, 2012), in contrast to a small condition in Deinonychus (Ostrom, 1969) and Dromaeosaurus (Colbert & Russell, 1969).

Figure 13 CT-rendered left (A, B, C) and right (D, E) mandibles of PMOL-AD00102, and a cross-sectional CT image of surangular in (F).

(A, D) in lateral views; (B) in dorsal view; (C, E) in medial views. Study sites: af, adductor fossa; an, angular; ar, articular; asf, anterior surangular foramen; co, coronoid; d, dentary; emf, external mandibular fenestra; imf, internal mandibular fenestra; pra, prearticular; saf, surangular foramen; sd, supradentary; sp, splenial; su, surangular; vcp, vertical columnar process.

Dentary

Both dentaries are preserved with the posterior end that bears the last four teeth (Fig. 13). As in Urbacodon (Alexander & Sues, 2007), the labial side of the alveoli is higher than the lingual side (Fig. 13E). The dentary bears a lateral groove as in other troodontids (Makovicky & Norell, 2004). As in Daliansaurus (Shen et al., 2017a), this dentary groove reaches the posterior end of the dentary. The posteroventral portion of the dentary is deep and sheet-like. The intramandibular process of the dentary is partially preserved and overlaps the anterodorsal surface of the anterior process of the surangular (Figs. 13A and 13D). At the ventral part of the intramandibular process, a small prong articulates dorsally with the small ventral groove of the anterior process of the surangular. The posteroventral part of the dentary overlaps the smooth lateral surface of the anterodorsal ramus of the angular with a broad, oblique suture. Medially, the dentary is overlapped by the splenial and the supradentary. A deep meckelian fossa is present between the dentary and the splenial. A deep socket for accommodating the surangular lies dorsal to the meckelian fossa.

Angular

Both angulars are nearly completely preserved (Fig. 13). The angular is bow-like, forming most of the ventral margin of the mandible posterior to the dentary. It forms the anterior and ventral borders of the external mandibular fenestra laterally, and the ventral border of the internal mandibular fenestra medially. Anteriorly, the angular upturns and articulates with the posteroventral part of the dentary and the splenial. Laterally, the angular extends posteriorly to the level of the surangular foramen and overlaps the surangular along a nearly straight suture posterior to the external mandibular fenestra (Figs. 2 and 13A). Medially, the angular forms the ventral border of the mandibular fossa, with the lateral wall slightly higher than the medial wall (Figs. 13C and 13E).

Surangular

The left and right surangulars are nearly completely preserved (Fig. 13). The surangular forms most of the dorsal margin of the mandible posterior to the dentary. Anteriorly, the surangular is straight and blade-like, and forms the dorsal border of the external mandibular fenestra. Its anterior end is blunt and wedged between the dentary and the coronoid. At the level of the midpoint of the external mandibular fenestra, a small anterior surangular foramen opens laterally, and extends as a groove anteriorly (Fig. 13A). Posterior to the foramen, the surangular is laterally swollen. The surangular becomes dorsoventrally deep posterior to the external mandibular fenestra, about twice as deep as the anterior portion and has a well-developed laterodorsal ridge. Medial to the laterodorsal ridge, a flat medial shelf of the surangular forms the dorsal border of the adductor fossa, making the cross section of the surangular “T”-shaped (Fig. 13F) as in the holotype (Xu, 2002; Xu et al., 2002), which is also a diagnostic feature of S. changii. Ventral to this laterodorsal ridge, a prominent surangular foramen is present laterally (Figs. 13A and 13D). Its diameter is about 30% of the depth of the posterior surangular, relatively larger than that in the holotype (Xu, 2002) and Gobivenator (Tsuihiji et al., 2014). The surangular is overlapped by the angular along a longitudinal suture ventrally.

Articular

Both articulars are preserved. The articular is semi-co-ossified with the prearticular medially and the surangular laterally. It bears two fossae that are separated by a rounded anteromedially-oriented ridge (Fig. 13B) to accommodate the quadrate condyles. The lateral fossa is shallower than the medial one. The mandibular fossae are more ventrally positioned than the dorsal margin of the mandible. Dorsally, the stout retroarticular process is sculptured by a transverse and deep groove anteriorly. This groove is probably for the attachment of the depressor mandibulae muscle, unlike Gobivenator in which this attachment is represented by a broadly concave surface (Tsuihiji et al., 2014). On the posteromedial margin of the retroarticular process, a vertical columnar process (Figs. 13B and 13C) is present as in dromaeosaurids (Currie, 1995).

Prearticular

The right prearticular is more completely preserved than the left one at the medial side of the postdentary portion (Fig. 13E). Anteriorly, the prearticular is deep and sheet-like, and forms the medial wall of the mandibular adductor fossa with the coronoid and the splenial. Anteroventrally, the prearticular encloses the internal mandibular fenestra with the angular (Fig. 13E). The internal mandibular fenestra is roughly crescentic, unlike the sub-rectangular internal mandibular fenestra in Dromaeosaurus (see Fig. 7E in Currie, 1995). Posterior to the internal mandibular fenestra, the ventral surface of the prearticular becomes mediolaterally wide and forms most of the ventral margin of the adductor fossa (Fig. 13C). More ventrally, the prearticular articulates with the angular. Posterolaterally, a trough is developed, and gradually slopes posteriorly. This trough is dorsally defined by a bony sheet whose anterior portion directs lateroventrally and the posterior portion directs laterodorsally. Medially, the prearticular overlaps the medial surface of the articular.

Splenial

Both splenials are partially preserved. The splenial anterior to the level of the last third dentary tooth is missing. The posterior margin of the splenial is forked on the medial side (Fig. 13E), but the posterodorsal branch of the left splenial is damaged (Fig. 13C). The posterodorsal branch gradually slopes down and contacts the medial surface of the coronoid and the prearticular. The posteroventral branch wraps the medial and ventral surfaces of the angular, and is laterally exposed as a broad triangle, as in other deinonychosaurians (Currie, 1995). Anterior to the contact with the angular, the splenial is shelf-like, and contacts the medial surface of dentary.

Coronoid and supradentary

The coronoid and the supradentary are preserved in PMOL-AD00102 (Fig. 13). In medial view, the strap-like supradentary overlaps the dentary immediately ventral to the alveolar margin. As in other non-avian theropods (Currie, 2003), the supradentary is co-ossified with the coronoid posteriorly. The coronoid is shelf-like and more than four times as deep as the supradentary (Fig. 13C). The ventral and dorsal margins of the coronoid are nearly parallel and the posterior half of the coronoid is concave medially. The posterior margin of the coronoid is bifurcated, forming the anterodorsal margin of the addcutor fossa (Fig. 13E). The dorsal process is slightly longer than the ventral one.

Dentition

Only the roots of the last two maxillary teeth are preserved on the left maxilla. The maxillary tooth row reaches close to the posterior end of the maxilla, like in other Jehol troodontids but different from Late Cretaceous troodontids.

The last four dentary teeth are preserved in each dentary (Fig. 13). The anterior two of these teeth are preserved with their crowns, and the last two teeth are nearly complete and located in alveoli. The alveoli are separated by a septa. The teeth are mediolaterally compressed. The crown curves posteriorly and its lateral and medial surfaces are flat. The medial carina is smooth, while the distal carina is serrated, as in the holotype (Xu, 2002), Sinusonasus (Xu & Wang, 2004), Daliansaurus (Shen et al., 2017a), Liaoningvenator (Shen et al., 2017b), Jianianhualong (Xu et al., 2017), Troodon (Currie, 1987b), Linhevenator (Xu et al., 2011), Sinornithoides (Currie & Dong, 2001), Saurornithoides and Zanabazar (Norell et al., 2009), in contrast to Xixiasaurus (Lü et al., 2010), Jinfengopteryx (Ji & Ji, 2007), Byronosaurus (Norell, Makovicky & Clark, 2000), Gobivenator (Tsuihiji et al., 2014), Almas (Pei et al., 2017a) and Urbacodon (Alexander & Sues, 2007), in which all teeth are unserrated. As in other troodontids (Makovicky & Norell, 2004), a constriction exists between the tooth crown and root.

On the right dentary, the third tooth from the last seems to be the largest among the preserved teeth with a height of the crown up to 3.7 mm. The second last alveolus bears a small replacement tooth that only has the crown tip exposed medially (Figs. 13C and 13E). The crown of the last tooth is half as high as its root.

Cervical Vertebrae

The paired proatlases and the anterior six cervical vertebrae are preserved in articulation (Fig. 14). The neural spines of the post-axis cervical vertebrae are broken more or less. The sixth cervical vertebra is only preserved with two prezygapophyses. The neural arch and the centrum are fused in post-atlas cervical vertebrae, implying that PMOL-AD00102 is an adult individual.

Figure 14 Cervical vertebrae of PMOL-AD00102.

(A) Photograph; (B) line drawing. Study sites: atic, atlantal intercentrum; atna, atlantal neural arch; atr, atlantal rib; ax, axis; c3-c6, third through sixth cervical vertebrae; di, diapophysis; ep, epipophysis; pa, parapophysis; pl, pleurocoel; pro, proatlas; r3-r5, third through fifth cervical ribs.

Proatlas

Both proatlas are well preserved in this specimen (Figs. 3 and 14). The proatlas is comprised of a main body and a posterior process. The posteroventral margin of the proatlas is curved (Fig. 15E). Medially, the proatlas has a concave surface (Fig. 15F). In lateral view, the main body is triangular and possibly articulates with the exoccipital anteriorly in life. The posterior process is thicker than the main body and is attached on the atlantal vertebral arch. The proatlas has only been reported in Gobivenator (Tsuihiji et al., 2014) among troodontids, but commonly exists in amniotes.

Figure 15 Selected CT-rendered cervical vertebrae of PMOL-AD00102.

Atlantal intercentrum in anterior (A), dorsal (B), posterior (C) and ventral (D) views; left proatlas in lateral (E) and medial (F) views; axis, axial rib and atlantal ribs in left lateral view (G); right atlantal neural arch in lateral (H) and medial (I) views. Study sites: amp, ampullae; ara, atlantal rib articulation; atr, atlantal rib; axi, axial intercentrum; axr, axial rib; ep, epipophysis; od, odontoid; ped, pedicle; pp, posterior process of proatlas; prz, prezygapophysis.

Atlas

The atlas is comprised of a centrum, an intercentrum and two neural arches. The atlantal arches and intercentrum are not fused in this specimen, as in dromaeosaurids and Aves (Norell & Makovicky, 2004). The atlantal centrum, namely odontoid, is co-ossified with the axis (Fig. 15I). The odontoid contacts the occipital condyle anteriorly and is positioned on the dorsal surface of the atlantal intercentrum. The odontoid is sub-coniform and wider than high in anterior view.

The atlantal intercentrum is U-shaped in anterior view (Fig. 15A). Anteroventral to the anterior end of the odontoid, a fossa defined by a septa on the intercentrum is developed to accommodate the occipital condyle (Fig. 15B). As in other non-avian theropods, this structure allows the skull to mobile up and down (Sereno & Novas, 1993). The articular surface with the atlantal arch on the atlantal intercentrum faces anteroventrally (Fig. 15C). The lateral edge of the posterior surface of the atlantal intercentrum is marked by a lip-like margin that is for the attachment of the capsular ligament as in Deinonychus (Ostrom, 1969). Ventrally, a facet on the posteroventral atlantal intercentrum is present, possibly for contacting the single-headed atlantal rib (Fig. 15D).

The paired neural arches are not co-ossified. The atlantal neural arch is triradiate with a stout postzygopophysis that articulates the lateral surface of the axis (Fig. 14). The epipophysis is present lateral to the zygopophyseal facet (Figs. 15G and 15H). At the base of each neural arch, the pedicle is slightly expanded in lateral view (Fig. 15G). The ampullae is tab-like and curves medially (Fig. 15G).

Axis

The axis is completely preserved, but broken into two parts (Fig. 14). The anterior part was scanned by CT with the skull and mandibles, as seen in Fig. 15I. The posterior part and the succeeding postaxial cervicals are shown in Fig. 14. The axis is well ossified, lacking the suture of the neural arch and the centrum. Anteriorly, the axis is co-ossified with the atlantal centrum as a well developed odontoid (Fig. 15I). Similarly, the axial intercentrum is co-ossified at the anteroventral corner of the axis. The intercentrum is short, about one fifth of the centrum in length. The intercentrum inclines anteroventrally, and forms a concavity for the atlantal intercentrum. This articulated structure is possibly functional for the lateral movement and rotation of the skull (Sereno & Novas, 1993).

The axial centrum is compressed bilaterally, and marked by two pleurocoels on each side (Fig. 14). The larger pleurocoel is centrally positioned, while the smaller one is dorsal to the former. Posteriorly, the centrum extends slightly beyond the neural arch, different from the condition in dromaeosaurids (Turner, Makovicky & Norell, 2012). The diapophysis and parapophysis are obscure by a slender axial rib that is preserved in articulation (Fig. 15I). Dorsally, the neural arch has a large neural spine. The neural spine is blade-like, and roughly triangular in lateral view. The dorsal margin of the neural spine is oblique posteriorly, and the posterior edge of the neural spine is almost vertical. Unlike Jianianhualong (Xu et al., 2017), the neural spine does not have a strongly posterodorsal expansion. Anteriorly, the prezygapophysis is small and extends anteroventrally beyond the odontoidal base slightly, as in Deinonychus (Ostrom, 1969). The postzygopophysis faces posteroventrally. The epipophysis is well developed (Fig. 14), nearly overlapping the entire postzygapophysis as in Byronosaurus (Norell, Makovicky & Clark, 2000). Posteriorly, the epipophysis is not beyond the postzygapophysis, contrary to the condition in some dromaeosaurids (Norell et al., 2006).

Postaxial cervical vertebrae

Four postaxial cervical vertebrae are preserved in articulation (Fig. 14). The articular facet between the adjacent cervical vertebrae inclines anteriorly, as in Oviraptorosauria and other Paraves (Turner, Makovicky & Norell, 2012). These vertebrae are comparable in size. The centrum extends posteriorly beyond the posterior margin of the neural arch, different from dromaeosaurids in which the centrum does not extend beyond the posterior end of neural arch (Turner, Makovicky & Norell, 2012). Dorsally, the centrum is fused with the neural arch. The sizes of the diapophysis and the prezygapophysis appear to increase gradually in the succeeding vertebrae. In contrast, the size of the epipophysis reduces posteriorly along the cervical series (Fig. 14).

The lateral surface of the third cervical vertebra is marked by two pleurocoels and a deep depression (Fig. 14). These two pleurocoels are located posteroventral and posterior to the diapophysis respectively. A deep depression is positioned more posteroventrally than the pleurocoels. The diapophysis and parapophysis are well separated (Fig. 14). The diapophysis is slender with a tongue-like shape in dorsolateral view. The articular facet of the diapophysis is smaller than that of the parapophysis. The articular facet of the prezygapophysis slopes anteroventrally. The postzygapophysis extends more laterally than posteriorly in dorsal view.

Cervical ribs

Two atlantal ribs, firstly reported in troodontids, are partially preserved lateroventral to the axial centrum in PMOL-AD00102. The atlantal rib is single-headed and curves ventrally (Fig. 15I), as in Archaeopteryx (Tsuihiji, 2017). The axial rib is more robust than the atlantal rib (Fig. 15I). The third cervical ribs are associated with the third cervical vertebra (Fig. 14). They are slender, and longer than the corresponding cervical centrum. The fourth and fifth ribs become more robust than the anterior cervical ribs.

Phylogenetic Analysis

In this study, we supplemented the phylogenetic dataset for coelurosaurians published by Xu et al. (2015b) with new anatomical information of PMOL-AD00102 and Sinusonasus. Two separate phylogenetic analyses were conducted. We treated PMOL-AD00102 as an independent terminal in the first analysis (92 terminals, 374 characters), and merged new codings of PMOL-AD00102 into the existing S. changii terminal in the second analyses (91 terminals, 374 characters). We added one additional state each for Character 6 and Character 8 to reflect the intermediate state of the subotic recess and the pneumatic lateral depression in IVPP V12615 and PMOL-AD00102 (see in the discussion section and the appendix for details). Phylogenetic analyses were performed with T.N.T. (Version 1.5; Goloboff, Farris & Nixon, 2015). Each analysis was run using the traditional search strategy with 1,000 replications, TBR and holding 10 trees per replication.

The first analysis produced 40 most parsimonious trees (MPTs) with a length of 1,433 steps (CI = 0.317, RI = 0.739). In the strict consensus topology, Mei was recovered as the basalmost troodontid, and PMOL-AD00102 was recovered in a polytomy with Sinovenator, Sinusonasus and the clade of other troodontids (Fig. 16A). This result does not recover PMOL-AD00102 and the original S. changii terminal as sister-group, because the dataset does not sample autapomorphies of S. changii (i.e., synapomorphies of IVPP V12615 and PMOL-AD00102). However, this analysis does recover PMOL-AD00102, Sinovenator and Sinusonasus at a similar “evolutionary stage” as we expected. To investigate the exact relationships of Jehol troodonitds requires a comprehensive and careful study of each taxon such as Sinovenator, Mei, Sinusonasus, Jinfengopteryx, Daliansaurus, Liaoningvenator and Jianianhualong, which is beyond the scope of this study.

Figure 16 Phylogenetic reconstruction.

(A) Troodontid portion of the strict consensus of 40 MPTs (TL = 1,433 steps, CI = 0.317, RI = 0.739), showing phylogenetic positions of Sinovenator and PMOL-AD00102; (B) Troodontid portion of the strict consensus of 20 MPTs (TL = 1,428 steps, CI = 0.318, RI = 0.739), showing phylogenetic position of Sinovenator.

After merging new codings of PMOL-AD00102 into the existing S. changii terminal, the second analysis produced 20 MPTs with a tree length of 1,428 steps (CI = 0.318, RI = 0.739). In the strict consensus topology, Mei was recovered as the basalmost troodontid while Sinovenator was recovered as the second basalmost troodontid that is more derived than Mei but less derived than Sinusonasus (Fig. 16B), which is similar to the result by Xu et al. (2017).

In both strict consensus topologies, Troodon, Zanabazar and Saurornithoides form a polytomy, and this clade instead forms a polytomy with IGM 100/44, Sinornithoides and Byronosaurus, as recovered by Xu et al. (2015b).

Discussion

Identification of PMOL-AD00102 as Sinovenator changii and comparisons with other Jehol troodontids

PMOL-AD00102 can be assigned to the Troodontidae based on the combination of individual characters that are typical of troodontids and/or have been regarded as synapomorphies for troodontids in different studies (Makovicky & Norell, 2004; Xu et al., 2017): a row of foramina along a longtitudinal line on the nasal; a well-developed supraorbital crest that expands laterally anterodorsal to the orbit on the lacrimal; a lateral ridge close to the ventral edge of the jugal; a pit on the ventral surface of the laterosphenoid; a reduced basal tubera that lie directly ventral to the occipital condyle; an oval-shaped foramen magnum; the quadrate bears a pneumatic fenestra and a lateral groove on the dentary.

We refer PMOL-AD00102 to S. changii based on the presence of a surangular with a “T”-shaped cross-section, even though our phylogenentic analysis does not resolve the relationships between PMOL-AD00102 and other specimens of S. changii, due to the lack of S. changii autapomorphies in the phylogenetic dataset. This diagnostic feature of S. changii (“T”-shaped cross-section of the surangular) was not reported in other newly discovered troodontid specimens and therefore supports the affiliation of PMOL-AD00102 to S. changii.

Another diagnostic feature suggested for S. changii (Xu et al., 2002), the antorbital fenestra with a vertical anterior margin is also present in the new specimen PMOL-AD00102. However, this feature alone could not refer PMOL-AD00102 to S. changii because a vertical anterior margin is also found in the antorbital fenestra of Sinusonasus.

PMOL-AD00102 is different from the type specimen of S. changii (IVPP V12615) in several other features: the frontal without the vertical lamina bordering the lacrimal, the presence of a septum between the basal tubera, the presence of a basisphenoid recess, a deep sagittal crest and the basipterygoid process with a blunt distal end. It is difficult to ascertain whether these differences are allometric, ontogenetic or preservational. The variation on the frontal-lacrimal contact is possibly preservational and the variation on the basal tubera could be preservational or ontogenetic, as we mentioned earlier in the description section. The variations of the basisphenoid recess and the distal end of the basipterygoid process could also be preservational, given the fragmentary nature of the holotype. The difference between the deep sagittal crest in PMOL-AD00102 and the shallow sagittal crest in the holotype could either be preservational considering the fragmentary parietal of the holotype, or be ontogenetic considering the sizes of both specimens. Regardless of these variations, we still attribute PMOL-AD00102 to S. changii instead of erecting a new taxon or attributing it to other existing troodontid taxa, until more fossil materials are available or more comprehensive studies are conducted.

Six other troodontids have been erected in the Jehol Biota: Mei, Sinusonasus, Jinfengopteryx, Daliansaurus, Liaoningvenator and Jianianhualong. Within these Jehol troodontids, the dentary teeth of Jinfengopteryx completely lack serrations (Ji & Ji, 2007). In contrast, other Jehol troodontids (except for Mei, unknown to the dentary teeth) including PMOL-AD00102 have serrated dentary teeth. PMOL-AD00102 seems different from Daliansaurus (see Figs. 2 and 3 in Shen et al., 2017a) by having the dentary with a small prong dorsally articulated with the surangular. PMOL-AD00102 differs from Mei (see Figs. 2A and 2B in Xu & Norell, 2004) and Sinusonasus (see Figs. 1 and 2 in Xu & Wang, 2004) by possessing the lacrimal with a bifurcated posterior process. PMOL-AD00102 also differs from Mei (Xu & Norell, 2004), Liaoningvenator (Shen et al., 2017b) and Jianianhualong (Xu et al., 2017) by the presence of a notched postorbital process of the frontal and a high and lamina-like saggital crest. PMOL-AD00102 can also be distinguished from Jianianhualong (Xu et al., 2017) by the presence of a mediolaterally narrow space between the jugal process and the pterygoid process of the ectopterygoid, the presence of an anterior surangular foramen, the surangular lacking a distinct fossa on its dorsal surface closed to its posterior end, the splenial with a forked posterior margin and the posterodorsal portion of the axial neural spine without a distinct posterior expansion.

Braincase of PMOL-AD00102 and Sinovenator changii

Sinovenator changii is the first troodontid reported from the Jehol Biota, and it was regarded as the most basal troodontid that has intermediate morphologies linking the two branches of deinonychosaurians: troodontids and dromaeosaurids (Xu et al., 2002). Sinovenator has typical deinonychosaurian features that are also observed in dromaeosaurids but absent from more derived non-Jehol troodontids, such as the non-arctometatarsalian pes, the opisthopubic condition of the pelvis, etc. Among these deinonychosaurian features, a primitive profile of the braincase (e.g., absence of the lateral depression, absence of the subotic recess, etc.) was suggested as key evidence that sets S. changii aside from more derived troodontids, in which the braincase has a well-defined lateral depression and a fully developed subotic recess. Although later reported troodontids from the Jehol Biota (Mei, Jinfengopteryx, Sinusonasus, Jianianhualong, Daliansaurus and Liaoningvenator) are also considered relatively primitive compared with their Late Cretaceous kins, no detailed morphologies of the braincase have ever been reported to prove/disprove this primitive condition of the braincase in Sinovenator and/or other Jehol troodontids. PMOL-AD00102, however, has a well-preserved cranial skeleton and provides a rare opportunity to investigate the early evolutionary trend of these morphologies in the troodontid braincase. Unlike reported in IVPP V12615, the new specimen PMOL-AD00102 shows a clear presence of the subotic recess, the otosphenoidal crest and basisphenoid recess.

The subotic recess is incipient in PMOL-AD00102 as a shallow depression, unlike the deep and clearly defined recess in Saurornithoides, Zanabazar, Troodon, and Byronosaurus. Although a typical subotic recess was not reported from the holotype of S. changii, that specimen (IVPP V12615) does have a lateroventrally faced depression lateroventral to the mid ear cavity and posterodorsal to the basipterygoid process (see Fig. 1B in Xu et al., 2002). This depression is located at the same position of the subotic recess of PMOL-AD00102 and derived troodontids. In contrast, such a structure is absent in dromaeosaurids and avialans (Norell et al., 2006). Here we regard these structures are homologous in IVPP V12615, PMOL-AD00102 and more derived troodontids, and the shallow subotic depression (incipient subotic recess) in Sinovenator represents an initial stage of the well developed subotic recess in more derived troodontids.

The otosphenoidal crest is present in PMOL-AD00102, although it is not as well developed as in the Late Cretaceous Troodon, Saurornithoides and Zanabazar. Typically, the otosphenoidal crest defines a lateral depression that hosts pneumatic cavities (e.g., the middle ear cavity and the subotic recess) on the lateral side of the braincase in Troodon, Saurornithoides and Zanabazar. The otosphenoidal crest in PMOL-AD00102 is more similar to that in Byronosaurus and Almas, in which the crest is positioned ventral to the opening for the facial nerve (CN VII) and dorsal to the anterior tympanic recess. A homologous structure also seems present in the braincase of IVPP V12615, at the same position between CN VII and the anterior tympanic recess (see Fig. 1B in Xu et al., 2002). This structure of IVPP V12615 seems more smooth and shorter than the otosphenoidal crest in PMOL-AD00102, but this difference is possibly preservational, as the braincase of IVPP V12615 undergoes a slight deformation and somewhat erosion. Therefore, we regard both PMOL-AD00102 and IVPP V12615 have an otosphenoidal crest that is not as developed as in Troodon, Saurornithoides and Zanabazar. The lateral depression defined by the otosphenoidal crest in these two specimens is not as developed as in Troodon, Saurornithoides and Zanabazar, either, but it resembles that in Byronosaurus and Almas, in which the mid ear region and CN VII fall outside of the lateral depression. Notably, the otosphenoidal crest in Sinovenator, Byronosaurus and Almas may be homologous to another curved ridge in Saurornithoides between CN VII and the anterior tympanic recess (see Fig. 11A in Norell et al., 2009). This curved ridge is ventral to the otosphenoidal crest in Saurornithoides, and therefore whether the so-called otosphenoidal crest in Sinovenator, Byronosaurus and Almas is homologous to that in Troodon, Saurornithoides and Zanabazar is unclear and needs more careful investigations.

The basisphenoid recess is a primitive character in coelurosaurians, and is observed in dromaeosaurids, Archaeopteryx and Anchiornis (Turner, Makovicky & Norell, 2012; Rauhut, 2014; Pei et al., 2017b). But the basisphenoid recess was thought to be lost in troodontids (Makovicky & Norell, 2004). Presence of the basisphenoid recess in the new specimen indicates that this morphology is possibly plesiomorphic in troodontids (at least present in the basal members, such as Sinovenator). In addition, the weakly-developed basisphenoid recess in Sinovenator possibly represents the initial stage of losing this recess in derived troodontids.

As discussed above, the braincase of Sinovenator is not as primitive as previously thought to be, although it still shows an intermediate profile between derived troodontids and non-troodontid paravians.

Notable new morphologies observed in PMOL-AD00102

Dromaeosaurids are characterized by the inverted “T”-shaped quadratojugal that contacts the lateral process and mandibular condyle of the quadrate and defines a large quadrate foramen (Norell et al., 2006). As a contrast, the quadratojugal is L-shaped and the quadrate does not have a lateral process in troodontids, and these features are regarded as plesiomorphies in non-dromaeosaurid paravians such as troodontids. However, because of the sparseness of well-preserved or well-exposed materials in troodontids, how exactly the quadratojugal articulates with the quadrate is unclear in this family. Fortunately, the quadratojugal and quadrate are well preserved in PMOL-AD00102, providing a rare opportunity to decipher the articulation of these two bones. The main body of the quadratojugal in PMOL-AD00102 overlaps the lateral surface of the lateral condyle of the quadrate as observed in Gobivenator (see Figs. 3A and 3C in Tsuihiji et al., 2014), and the squamosal process of the quadratojugal in PMOL-AD00102 wraps the posterior surface of the quadrate as in Sinornithoides (Russell & Dong, 1993). The quadratojugal wraps the lateral and the posterior surfaces of the quadrate in troodontids, unlike the condition in oviraptorids and dromaeosaurids in which the quadratojugal is articulated with the quadrate only on the lateral side (Osmólska, Currie & Barsbold, 2004; Norell et al., 2006). Thus, this quadrate-quadratojugal articulation in troodontids is different from that in oviraptorids and dromaeosaurids, and probably represents an apomorphy related to the feeding styles in the Troodontidae.

The stapes is a delicate bone, and rarely preserved in non-avian coelurosaurians. To date, the stapes was only found in dromaeosaurids, oviraptorids and tyrannosaurids (Colbert & Ostrom, 1958; Clark, Norell & Rowe, 2002; Witmer & Ridgely, 2009), but the stapes in these findings are either incomplete or have only been briefly mentioned. Here, as the first report in troodontids, the stapes of PMOL-AD00102 are well revealed by using the CT-scan technique. The stapes of PMOL-AD00102 directs both posterolaterally and ventrally, and positioned outside a groove in the paroccipital process, as in tyrannosaurids and oviraptorids, but in contrast to the posterolaterally directed stapes that hosted in a groove along the paroccipital process in dromaeosaurids. Therefore, PMOL-AD00102 seems to have a conservative way of structuring the otic bone like in more primitive coelurosaurians but unlike the more closely related dromaeosaurids. In addition, the stapes in PMOL-AD00102 firstly reveals some new morphological information on the ear of non-avian coelurosaurians, such as the subtriangular footplate and the posteriorly inclined stapedial shaft. As far as we know among dinosaurians, the shape of the footplate is nearly square in Allosauroidea (Madsen, 1976), semicircular in Sauropodomorpha (Chapelle & Choiniere, 2018) and unknown in Ornithischia. Therefore, even though the stapes is commonly present in dinosaurians, the morphology of the footplate varies in different lineages.

The epipterygoid was hypothesized to be lost in all troodontids by Tsuihiji et al. (2014) based on a previous study of Gobivenator. However, our observation with the new specimen shows the epipterygoid is actually present in Sinovenator, as firstly reported in the Troodontidae. This implies that the loss of the epipterygoid is likely a derived character that present in later diverging taxa of the family. Moreover, this finding supports the hypothesis that the loss of the epipterygoid is possibly homoplastic in derived troodontids and avialans (except for Archaeopteryx) (Tsuihiji et al., 2014).

The atlantal ribs have never been reported in troodontids due to the rare preservation of the elements. The atlantal ribs are well preserved in PMOL-AD00102 and have a slender shape, which supports the hypothesis that the atlantal rib has an evolutionary trend to reduce the size along the theropod lineage (Tsuihiji, 2017). Additionally, the troodontid atlantal rib curves ventrally as in basal birds (Tsuihiji, 2017), unlike the straight condition in dromaeosaurids (see Fig. 2 in Xu et al., 2010).

3D reconstruction based on the CT-scan data of PMOL-AD00102 reveals other characters that have not been noticed or rarely preserved in troodnotids, though these characters are more common in other paravians. A vertical columnar process on the articular and the preorbital bar of the lacrimal not contacting the maxilla is firstly reported in troodontids as observed in this new specimen. A vertical columnar process of the articular is a typical character only reported in dromaeosaurids (Currie, 1995), and the presence of this character in Sinovenator indicates it is probably plesiomorphic in deinonychosaurians and secondarily lost in derived troodontids. As in dromaeosaurids (Norell & Makovicky, 2004), Gobivenator (see Fig. 5 in Tsuihiji et al., 2014) and Archaeopteryx (Elzanowski, 2001), 3D reconstruction of the palate shows that the pterygopalatine fenestra is long in this new specimen, whereas this fenestra is small in ornithomimosaurs (Osmólska, Roniewicz & Barsbold, 1972) and therizinosaurs (Clark, Maryañska & Barsbold, 2004), and absent in oviraptorosaurs (Elzanowski, 1999) and other avialans (except for Archaeopteryx). Therefore, the long pterygopalatine fenestra is possibly plesiomorphic for Paraves in accordance with the conclusion that the pterygoid process of the palatine has an apparently lengthening trend toward the basal Avialae (Tsuihiji et al., 2014), and secondarily lost in derived avialans.

Conclusion

PMOL-AD00102, a new specimen referred to S. changii, is described in detail with the assistance of the CT-scan data. More cranial and cervical anatomies and diagnostic features of S. changii are revealed, such as a well-developed medial shelf on the jugal, a slender bar in the parasphenoid recess, a lateral groove on the pterygoid flange of the ectopterygoid, and the lateral surface of the anterior cervical vertebrae bearing two pneumatic foramina.

In addition, we find the braincase of S. changii is not as primitive as previously suggested, although it still shows an intermediate state between derived troodontids and non-troodontid paravians in having an initial stage of the subotic recess and the otosphenoidal crest.

Moreover, our new observation on PMOL-AD00102 has revealed several new and/or detailed anatomical information on the quadrate-quadratojugal articulation, the stapes, the epipterygoid, the atlantal ribs, etc.

Supplemental Information

Supplemental Information 1 Appendix–Updated information for phylogenetic analysis.

Click here for additional data file.

Supplemental Information 2 3D reconstruction of ct data.

Click here for additional data file.

We thank Prof. Ke-Qin Gao and Dr. Jia Jia (Peking University), Dr. Hong-Yu Yi (IVPP), and Mr. Qin-Fang Fang (China University of Geosciences) for their help in CT scan. We also thank the academic editor Dr. Hans-Dieter Sues and the three reviewers including Dr. Takanobu Tsuihiji, Dr. Mark Loewen, and one anonymous reviewer for their helpful comments and suggestions that greatly improved the quality of our manuscript.

Additional Information and Declarations

Competing Interests

Author Contributions

Data Availability

The authors declare that they have no competing interests.

Ya-Lei Yin conceived and designed the experiments, performed the experiments, analyzed the data, prepared figures and/or tables, authored or reviewed drafts of the paper, approved the final draft.

Rui Pei conceived and designed the experiments, performed the experiments, analyzed the data, authored or reviewed drafts of the paper, approved the final draft.

Chang-Fu Zhou conceived and designed the experiments, performed the experiments, contributed reagents/materials/analysis tools, prepared figures and/or tables, authored or reviewed drafts of the paper, approved the final draft.

The following information was supplied regarding data availability:

The raw data are provided in the Supplemental Files.

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
