# Peer review of "Cranial morphology of Sinovenator changii (Theropoda: Troodontidae) on the new material from the Yixian Formation of western Liaoning, China"

_PeerJ, doi:10.7717/peerj.4977_

## Round 0.1 · original submission · Minor Revisions

All reviewers agree that this is an excellent contribution. Please respond to the various comments and suggestions offered by the reviewers in preparing the revised version of the manuscript.

·

Basic reporting

Overall, the manuscript is very well-written. So far as I know, all relevant references are cited. The most commendable, greatest aspect of this manuscript is excellence of figures, especially individually segmented bones based on CT data. These figures will be invaluable references for future researches on theropod morphology.
One aspect of writing that I noticed is that the authors' descriptions sometimes contain seemingly unnecessary statements. For example, lines 391-395 states "The frontal contacts the nasal anteriorly, the lacrimal anterolaterally, the postorbital posterolaterally, the parietal posteriorly, and the laterosphenoid posteroventrally." I believe this condition is universal in Dinosauria and thus such a description seems to be unnecessary. Of course, it is not an incorrect statement and, considering this is an electronic journal in which you do not worry about the page number of the article, leaving these statements as they are is also permissible.

Experimental design

The analysis including digital segmentation and phylogenetic analyses are appropriately done. This manuscript provide so much of detailed information of cranial morphology of troodontids, which tend to be under-represented in the fossil record.

Validity of the findings

This is a solid and substantial description of osteology of a previsouly poorly-known taxon. So much new information is reported here. The conclusion is also well-supported by their observations.

Additional comments

I am really impressed by the quality of the manuscript, especially 3D-segmented images of bones and detailed description. I made some suggestions in writing, as well as a couple of questions, in the attached PDF. Because I am also a non-native English speaker, I do not claim all my suggestions would improve the quality of MS. However, I believe at least some of them are relevant. Anyway, I am very excited to read through MS, especially the description of the proatlas and atlas ribs!

Reviewer 2 ·

Basic reporting

The manuscript by Yin and colleagues provides a detailed anatomical description of cranial material, which is referred to the genus Sinovenator, a troodontid from the Lower Cretaceous of China. The manuscript is well structured and illustrated with many figures, which help to understand the description. Apart from a couple of typos, the English is very well. The relevant literature is cited and the study is presented in a broader context.

Experimental design

It is absolutely necesarry that more information on the discovery of the fossil is provided (see my comment below).

Validity of the findings

As mentioned above, the manuscript containes a detailed anatomical description, which needs only minor improvement. Both text and figures allow to understand the anatomy of the fossil and all conclusions drawn from the description are sound. I am with the authors, that the new material resembles Sinovenator best. However, due to the fragmentary nature of both the holotype and the referred material, only one diagnostic character was identified that allows a classification as Sinovenator. On the other side, the new material shows also some differences to Sinovenator, which are listed in the text. It would be good, if these differences could discussed in more detail, because they are relevant for the final classification as Sinovenator or maybe cf. Sinovenator. Furthermore, could the authors list more differences to Sinusonasus? Why was the latter taxon not included in the phylogeny? As Sinusonasus has an interesting mix of characters it would be relevant to know, where it is placed within the tree, also relative to Sinovenator and the new fossil.

Additional comments

- P7L34f: “As a close relative ...” - In my opinion, this sentence can be deleted to avoid repetitions.

- P8L62: Can the authors provide more information on the discovery of the fossil. When and where was the fossil found? Was it found during an excavation of a Chinese research institute?

- P8L69: Change “uA” to “μA”

- P9L79,L81,L83: Please, provide the name of the institutes abbreaviated as IVPP and PMOL so that it is known where the specimens are housed.

- P10L98: What makes the skull so remarkable?

- P10L101: Please mention that the anterior margin of the AOF is not complete and that the shape of area is an interpretation.

- P10L107: Please add that the ascending process of the maxilla is only preserved as a small peace of bone on the right side of the skull.

- P10L109: The contact between the ascending process of the maxilla and the anterior contact of the lacrimal can be better seen in Fig. 4B, may add.

- P10L109: Please, add that only the ventral portion of the interfenestral bar is preserved.

- P10L111: Please, add that the ventral ramus of the maxilla = jugal process

- P11L124: Change to “develops”.

- P12L146: Please add that the anterior process of the lacrimal points anteroventral.

- P14L182: Change “and” to “but”.

- P15L207: Change “The suborbital process” to “It” to avoid repetitions (see sentence before).

-P17L243: Please add, that the QJ covers the ventral portion of the quadrate laterally. As written on can assume it covers the quadrate of the whole dorsoventral dimension.

- P17L252: Change “The quadrate head” of the second sentence to “It” to avoid repetitions (see sentence before).

- P20L321ff: The seperation of the anterior triangular depression and a posterior sub-triangular depression by a transferse ridge can be also found in the Schamhaupten specimen of Archaeopteryx, so may cite Rauhut et al. (2018).

- P24L396f: May add further comparsions with other Jehol troodontids, like Mei.

- P38L694: Change “proatlas” to “proatlases”.

- P46L861: Non-arctometatarsalian pes is the normal condition in theropods, not only dromaeosaurids.

·

Basic reporting

No comment

Experimental design

No comment

Validity of the findings

No comment

Additional comments

Some of the strongest parts of the paper are the descriptions. I strongly recommend addressing with figures the referral to Sinovenator as I could just as easily believe you have a new taxon based on what has been figured. My photos of the type don’t include the surangular, so again I believe you are correct in your referral, I just cannot assess it for myself. Great paper and very nice illustrations. In conclusion, I recommend acceptance of the paper whether or not the concerns below are addressed.

Notes to the authors:

I enjoyed the paper, and I have just a few suggestions that I believe will make the paper stronger.

Again, just a little more on the referral would be nice.

General comments:

Abstract:
This is a good summary, I will have to take your word for the referral as the original description does not figure the surangular cross-section.

Introduction:
Lines 53-55: I will have to take your word for the referral as these features are not figured for the type.

Materials and Methods:
Line 64: replace miss with lack.
RESULTS

Locality, horizon, and AGE.
Line 85: add age to heading.

Revised Diagnosis
Very nice. Much better than what we have been going off of.

Description:
Line 98: The skull preserves a partial antorbital fenestra, large orbit and temporal fenestrae.
Line 191: Distally, this process is isolated from the quadrate shaft; likely due to taphonomic distortion.

Discussion:
Very nice work. I like that you discuss whether or not it is Sinovenator but are conservative. Every paper should be written in this way!

Figures
Overall the figures are exquisite.
Consider adding a cross-sectional image of the jugal and surangular

If you have any questions feel free to contact me at [email protected].

---

## Round 0.2 · accepted · Accept

The revised version of the manuscript is now recommended for acceptance for publication.

#